# Device Status Evaluation Method Based on Deep Learning for PHM Scenarios

**Pengjun Wang** [1,2,*]**, Jiahao Qin** [1]**, Jiucheng Li** [1]**, Meng Wu** [2]**, Shan Zhou** [2] **and Le Feng** [2]

[1] Department of Electronic Engineering, Tsinghua University, Beijing 100080, China
[2] Smartbow Tech., Inc., Beijing 100080, China
[*] Correspondence: wangpjz@mail.tsinghua.edu.cn

**Abstract:** The emergence of fault prediction and health management (PHM) technology has proposed a new solution and is suitable for implementing the functions of improving the intelligent management and control system. However, the research and application of the PHM model in the intelligent management and control system of electronic equipment are few at present, and there are many problems that need to be solved urgently in PHM technology itself. In order to solve such problems, this paper studies the application of the equipment-status-assessment method based on deep learning in PHM scenarios, in order to conduct in-depth research on the intelligent control system of electronic equipment. The experimental results in this paper show that the change in unimproved deep learning is very subtle before the performance change point, while improvements in deep learning increase the health value by about 10 times. Thus, improved deep learning amplifies subtle changes in health early in degradation and slows down mutations in health late at performance failure points. At the same time, comparing health-index-evaluation indicators, it can be concluded that although the monotonicity of the health index is low, its robustness and correlation are significantly improved. Additionally, it is very close to 1, making the health index curve more in line with traditional cognition and convenient for application. Therefore, an in-depth study of methods for health assessment by improving deep learning is of practical significance.

**Keywords:** PHM scene; deep learning; health status assessment; neural network

## 1. Introduction

The main functions of the current intelligent management and control system are status detection and abnormal alarm. However, the expected functions of the intelligent management and control system are by no means limited to status detection and alarming. The intelligent management and control system should include equipment status monitoring, equipment fault diagnoses, equipment predictive maintenance, and autonomous control. In the fields of aerospace, rail transit, power pipeline network, etc., that require high reliability and safety, the fault prediction and health management model (PHM) has been proposed by researchers. It is used to provide strategic operation and maintenance support, and it has successfully carried out engineering practice in complex systems, such as aerospace engines and ship equipment. Facts have proved that the PHM model can effectively reduce accidents, significantly reduce maintenance costs, significantly improve the reliability of equipment parts, and ensure the safe and stable operation of complex systems. This paper first analyzes the PHM health-status-assessment process and introduces relevant indicators of health status assessment. A health-state-assessment method based on improved deep learning is proposed. Finally, the health assessment of the PHM model is analyzed through simulation experiments. It aims to make certain contributions to the health assessments of the PHM model.

According to the research progress in foreign countries, different researchers have conducted corresponding cooperative research in PHM. Hamadache M reviewed the

basics of prognostic and health management (PHM) techniques for REB. Additionally, he discussed deep-learning methods for REB fault detection, diagnoses, and prediction [1]. Xia T studied the newly proposed PHM method. As a basis for decision making, he proposed an algorithm for predicting machine health based on operating load. At the machine level, he studied dynamic multi-attribute maintenance models for different machines in CPS [2]. Feng D proposed a new framework for TPSS maintenance using predictive and health Management and proactive maintenance (PHM-AM) techniques. He also described in detail the hardware components and software modules required to implement the proposed PHM-AM framework [3]. However, these scholars' research on PHM lacks technical proof. After this research, it was found that the research on PHM scenarios based on deep learning has a certain reliability, and we have checked the relevant literature on deep learning.

At present, scholars have conducted in-depth research on deep learning. Mater A C aimed to explain the concepts of deep learning to chemists from any background, followed by an overview of the various applications presented in the literature. The hope is to enable the wider chemical community to participate in this emerging field and to promote deep learning to accelerate the ongoing development of chemistry [4]. Min S reviewed deep learning in bioinformatics and presented examples of current research. To provide a useful and comprehensive perspective, Min S categorized studies by bioinformatics field and deep-learning architecture and presented a brief description of each study [5]. Deep-learning methods are a class of machine-learning techniques capable of identifying highly complex patterns in large datasets. Zou J provided a perspective and primer on deep-learning applications for genome analyses [6]. However, these scholars did not conduct research on the application of deep-learning-based device-state-assessment methods in PHM scenarios, but only discussed its significance unilaterally.

The innovations of this paper are as follows: First, it introduces the evaluation process of PHM health status. This paper proposes a health-state-assessment method based on improved deep learning, which is improved on the basis of traditional deep learning. It uses improved deep learning as the health index, constructs the health index curve, and intuitively and quantitatively evaluates the health status of complex equipment or systems. Finally, this paper discusses the health status assessment of the PHM model.

## 2. Method of Device State Based on Deep Learning in PHM Scene

### 2.1. PHM Health-Status-Assessment Process

Health status assessment is a key link in predictive maintenance. It plays a vital role in the PHM model of the entire intelligent management and control system. The research work of health status assessment is mainly focused on the assessment methods, which are mainly based on the characteristics of specific research objects and various assessment methods. The health-status-assessment process is slightly different according to the different objects controlled by the intelligent management and control system. However, for complex devices/systems, they can be roughly divided into two types according to their compositional structures: hierarchical structures and non-hierarchical structures [7]. Common methods of health status assessment include the following: model method; analytic hierarchy process; fuzzy evaluation method; artificial neural network method; Bayesian network-based method; grey theory; extension theory, etc. Figure 1 shows the PHM system.

A hierarchical structure refers to the design according to the level, from the basic components to the functional modules to the subsystems, and then combined into a whole device/system. A non-hierarchical structure means that the internal relationship of equipment/system is complex, and it is difficult to separate the layers [8].

For the health assessment of complex equipment/systems with a hierarchical structure, the complex/equipment system should be divided into several subsystems according to the hierarchy, and then the health assessment should be carried out according to the structure of the subsystems. These subsystems can be divided into three categories, and the first category of the subsystems can directly determine the key parts and key components. It

then obtains the characteristic parameter data of the key parts and the key devices in a targeted manner, and finally evaluates the health status of the subsystem. The second type of subsystem can also continue to be divided into layers. For this type of subsystem, it should always be divided into layers until the key parts and devices are determined, and then the health status assessment is carried out according to the process of the first type of system [9]. The third type of subsystem can directly obtain characteristic information for health assessment [10].

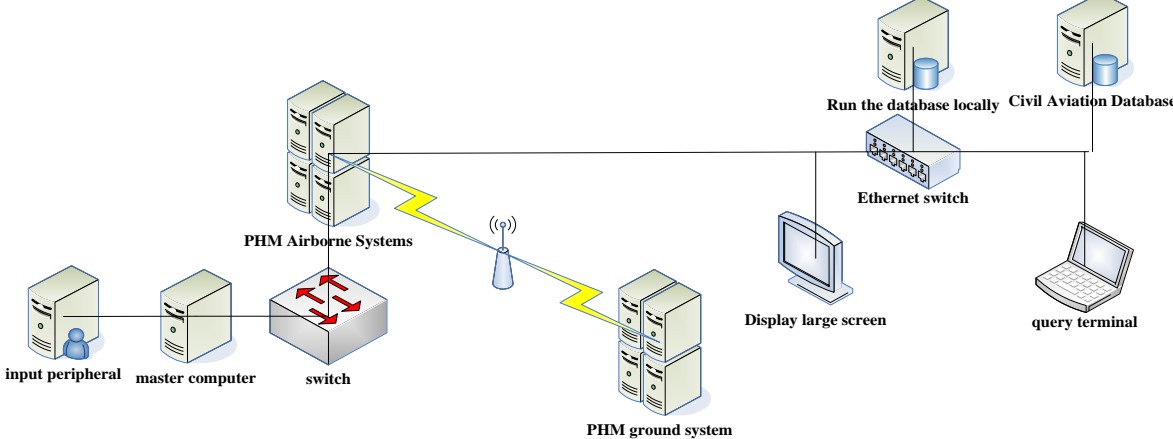

**Figure 1.** PHM system.

For non-hierarchical complex equipment/systems, it is difficult to divide subsystems. Therefore, it is necessary to select and extract characteristic parameters from operating data, experimental data, work logs, test data, etc., according to the knowledge and experience base. It then undergoes a health status assessment [11].

As shown in Figure 2, the main process of health status assessment consists of three steps: data processing; feature extraction; and the construction of a health index curve. Data processing refers to the cleaning and normalizing of the collected data so as to be more suitable for feature extraction [12]. Feature extraction refers to the use of signal analyses and processing techniques to extract various features from the data. It is worth noting that not all characteristics are beneficial to health status assessments. Therefore, how to extract features containing degraded information needs to be combined with knowledge and experience to make research decisions [13].

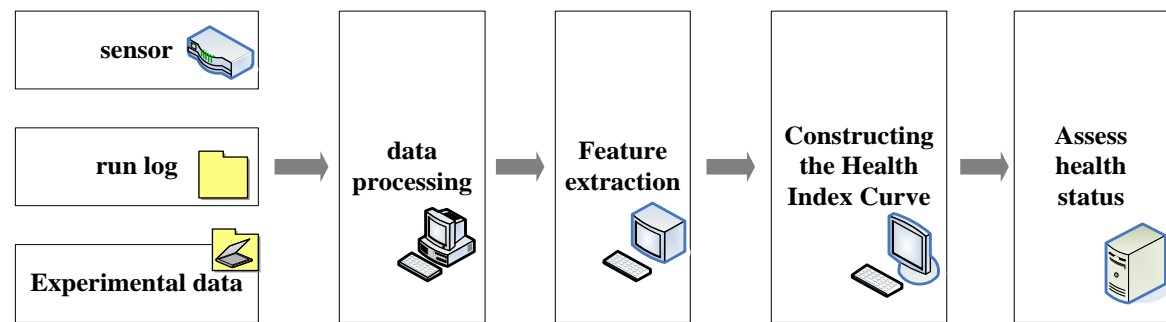

**Figure 2.** Health-status-assessment process.

### 2.2. Indicators Related to Health Status Assessment

Because the parameters constructed in the health status assessment establish a quantitative correspondence with the health status of complex equipment/systems, the constructed parameters often do not have actual physical meaning. Therefore, it is necessary to evaluate the health index curve to determine whether it has the ability to characterize the health status of complex equipment/systems [14]. There are three commonly used performance

evaluation indicators: monotonicity; robustness; and correlation. The relevant indicators are introduced in detail below.

(1)  Monotonicity

As a general rule, the degradation process of equipment/system performance should be monotonic, that is, the performance gradually decreases until it fails completely. Therefore, the health index curve constructed by the study should also be monotonic, and the corresponding value of its health status can be from high to low, or from low to high. It can also be changed within a certain interval through data transformation [15]. The most commonly used definition of monotonicity is as follows:

$$Mon(A) = \left| \frac{\sum_{p=1}^{H-1} \tau(a_p - a_{p+1}) - \sum_{p=1}^{H-1} \tau(a_{p+1} - a_p)}{H-1} \right| \tag{1}$$

In the formula, $Mon(A)$ represents the monotonicity strength; $A = \{a_1, a_2 \cdots a_h\}$ is the constructed health index sequence; $a_p$ is the health index at the $p$-th time; $H$ is the length of the health index sequence; and $\tau(a)$ is the unit step function, $a \geq 0$, $\tau(a) = 1$. When $a < 0$, the value of $Mon(A)$ is in the interval [0, 1]. The closer $Mon(A)$ is to 1, the better the monotonicity of the curve, and the worse the monotonicity trend is [16].

(2)  Robustness

Robustness is an evaluation index of stability, which is used here to quantify the anti-interference ability of the health index curve. That is to say, the higher the robustness of the stability under external interference, the smoother the corresponding health index curve should be. The definition of robustness is as follows:

$$Rob(A) = \frac{\sum_{p=1}^{H} exp\left(-\left|1 - a_{\tilde{p}} / a_p\right|\right)}{H} \tag{2}$$

In the formula, $Rob(A)$ represents the robustness value; and $a_{\tilde{p}}$ represents the health trend value at the $p$-th time. The value of $Rob(A)$ is also in the interval [0, 1]. The closer $Rob(A)$ is to 1, the better the robustness is. Otherwise, the robustness is poor [17].

(3)  Correlation

Correlation refers to the correlation between the health index curve and time/round. The higher the correlation, the more accurate the description of the degradation process. The definition of correlation is as follows:

$$Cor(A) = \left| \frac{H\left(\sum_{p=1}^{H} a_p r_p\right) - \left(\sum_{p=1}^{H} a_p\right) \sum_{p=1}^{H} r_p}{\sqrt{\left[H\left(\sum_{p=1}^{H} a_p{}^2\right)\left(\sum_{p=1}^{H} a_p\right)^2\right]\left[H\left(\sum_{p=1}^{H} r_p{}^2\right)\left(\sum_{p=1}^{H} r_p\right)^2\right]}} \right| \tag{3}$$

In the formula, $Cor(A)$ represents the correlation value; $r_p$ represents the $p$-th moment; the value of $Cor(A)$ is also in the interval [0, 1]. The closer the $Cor(A)$ is to 1, the stronger the correlation between the health index curve and time [18].

After determining the indicators of health status assessment, the health index curve is constructed from the characteristic data. This paper proposes improved deep learning as a health value and a method to construct a health index curve using linear regression model fitting [19].

### 2.3. Health-Status-Assessment Method Based on Improved Deep Learning

In machine learning, pattern recognition, and image processing, feature extraction begins with an initial set of measurement data and establishes derived values (features) aimed at providing information and non-redundancy, thus facilitating subsequent learning

and generalization steps, and in some cases bringing better interpretability. There are two main deficiencies in the current research on health status assessment. The key to making up for these two deficiencies lies in the selection of feature parameters and evaluation methods, but extracting features is an extremely challenging task. Most of the current methods for assessing health status rely heavily on feature selection, and most of the constructed health factors are for specific scenarios, which are difficult to use in other systems [20]. Therefore, this paper hopes to find a health-status-assessment method that can construct a suitable health factor through simple feature extraction. Therefore, this paper chooses to use deep learning to calculate the deviation between the current data and the initial healthy sample data. The calculated result is used as a health index, which evaluates the health status [21–23]. It improves on the basis of traditional deep learning, uses the improved deep learning as a health index, builds a health index curve, and intuitively and quantitatively evaluates the health status of complex equipment or systems.

### 2.4. Deep Neural Network Model and Training Algorithm

Deep learning has made many achievements in search technology, data mining, machine learning, machine translation, natural language processing, multimedia learning, voice, recommendation and personalized technology, and other related fields. A deep neural network model of deep learning is constructed based on deep-learning theory. The model extracts, processes, and fuzzes information through neurons in each layer. Then, the neurons of the multi-layer network perform nonlinear transformations on the data features. It makes data feature information more abstract by low-level features. Throughout the learning process, the network model can self-learn based on a large amount of data information. The distribution characteristics and complex function representations of learning information are stored in the deep neural network to make it have the ability to recognize and perceive [24–26]. The concept of deep learning originates from the research of artificial neural network. The multi-layer perceptron with multiple hidden layers is a good example of a deep-learning model. The accuracy rate and the recall rate are contradictory. The recall rate is often very low when the model has a high accuracy rate. On the contrary, the higher the recall rate is, the lower the accuracy rate is. Therefore, in practical applications, we often need to consider the accuracy rate, accuracy, and recall rate as a whole according to the actual needs so as to train the optimal model.

A deep neural network is a structure that contains multiple single hidden neural networks. As shown in Figure 3, the characteristic of such a network is that all node neurons of each level are interconnected with all node neurons of the next level. There are no neuron nodes in any layer that are not connected. The learning ability of the deep neural network is reflected in the expression of the weights and biases of each neuron, and it is used to fit a powerful mapping relationship. Different from traditional shallow learning, deep learning is different in that it emphasizes the depth of the model structure, which usually has 5, 6, or even 10 layers of hidden nodes. It clearly highlights the importance of feature learning, that is, through layer-by-layer feature transformation, the feature representation of the sample in the original space is transformed into a new feature space, which makes classification or prediction easier.

(1)  Network forward propagation

It is assumed that the input layer and output layer of a single hidden layer network contain $Q$ and $P$ neurons, respectively. There are $K$ neurons in the middle hidden layer, and the link weights from the input layer to the hidden layer and the hidden layer to the output layer are $u_{nm}$, $e_{hn}$, respectively. Then, in the hidden layer, the input of the nth neuron is:

$$net_n = \sum_{m=1}^{Q} u_{nm}a_m - \varphi_n \ (n = 1, 2, \ldots, K) \tag{4}$$

The output of the nth neuron in the hidden layer is:

$$r_n = g(net_n) \ (n = 1, 2, \ldots, K) \tag{5}$$

In the formula, $g(net_n)$ represents the activation function of the hidden layer neuron node, and the function selects the ReLU type as shown in the following formula.

$$g(a) = \begin{cases} a & a \geq 0 \\ 0 & a < 0 \end{cases} \tag{6}$$

The input received by the hth neuron in the output layer is:

$$net_h = \sum_{n=1}^{K} e_{hn} r_n - \varphi_h \ (h = 1, 2, \cdots, P) \tag{7}$$

The output is:

$$r_h = g(net_h) \ (h = 1, 2, \cdots, P) \tag{8}$$

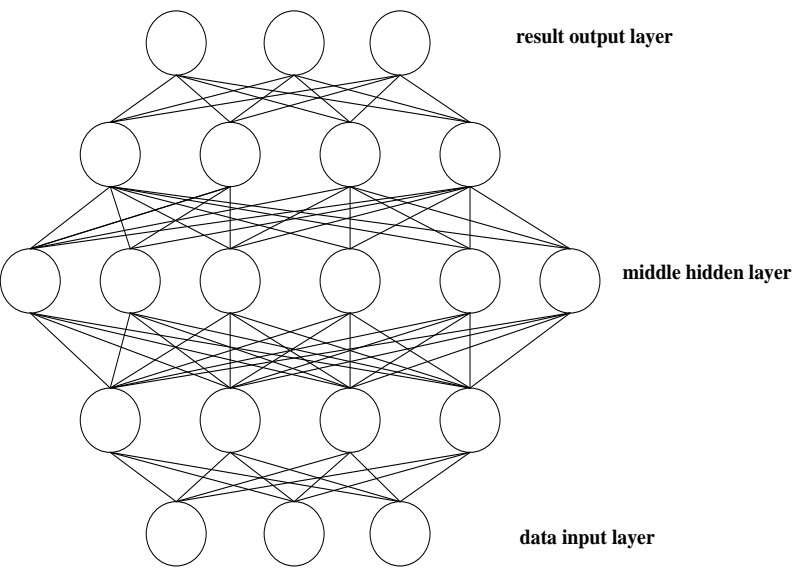

result output layer

middle hidden layer

data input layer

**Figure 3.** Deep neural network structure.

(2)    Network backpropagation

Backpropagation is to pass the error signal from the output layer to the input layer layer-by-layer and adjust the weight coefficients during the transmission process to make the network output reach the desired output. A backpropagation algorithm, referred to as a BP algorithm, is a learning algorithm suitable for multi-layer neural networks. It is based on the gradient descent method.

The output error for a single sample is:

$$M_i = \frac{1}{2} \sum_{h=1}^{P} (s_h - r_h)^2 \tag{9}$$

Then, the total error of the network for all samples $C$ is:

$$M = \sum_{i=1}^{C} M_i = \frac{1}{2} \sum_{i=1}^{C} \sum_{h=1}^{P} (s_h - r_h)^2 \tag{10}$$

The weight adjustment formula of the output layer is:

$$\Delta e_{hn} = -\sigma \frac{\partial M_i}{\partial e_{hn}} = -\sigma \frac{\partial M_i}{\partial net_h} \cdot r_n \tag{11}$$

$\sigma$ is the learning rate $0 < \sigma < 1$.

$$\gamma_h = -\frac{\partial M_i}{\partial net_h} = (r_h - s_h) \tag{12}$$

It can be obtained by

$$\Delta e_{hn} = \sigma \gamma_h r_n = \sigma(r_h - s_h)r_n \tag{13}$$

The weight adjustment formula of the hidden layer is:

$$\Delta u_{nm} = -\sigma \frac{\partial M_i}{\partial u_{nm}} = -\sigma \frac{\partial M_i}{\partial net_n} \cdot \frac{\partial net_n}{\partial u_{nm}} = \sigma \frac{\partial M_i}{\partial net_n} \cdot r_m \tag{14}$$

Let

$$\gamma_h = -\frac{\partial M_i}{\partial r_n} \tag{15}$$

Because the output of the hidden layer has a direct impact on the input of its connected layers, then:

$$-\frac{\partial M_i}{\partial r_n} = \sum_{n=1}^{K} \left(-\frac{\partial M_i}{\partial u_{nm}}\right) \cdot e_{hn} = \sum_{n=1}^{L} \gamma_h \cdot e_{hn} \tag{16}$$

So

$$\gamma_n = \sum_{n=1}^{L} \gamma_h \cdot e_{hn} \tag{17}$$

Combining the above formulas, we can get:

$$\Delta u_{nm} = \sigma \left(\sum_{n=1}^{L} \gamma_h \cdot e_{hn}\right) r_m \tag{18}$$

So, the update formula for any output layer weights and biases is:

$$e_{hn}(h+1) = e_{hn}(h) + \Delta e_{hn} = e_{hn}(h) + \sigma \gamma_h r_n \tag{19}$$

$$u_{nm}(h+1) = u_{nm}(h) + \Delta u_{nm} = u_{nm}(h) + \sigma \gamma_n r_m \tag{20}$$

## 3. Experimental Results of the Device State Based on Deep Learning in the PHM Scene

In order to verify the health assessment algorithm based on improved deep learning, this paper combines the composition characteristics of a new type of signal acquisition equipment. It takes the classic Sallen–Key low-pass filter circuit as an example to carry out simulation analyses to illustrate the health-status-assessment process of key circuits in electronic equipment. The circuit schematic of the Sallen–Key low-pass filter is shown in Figure 4:

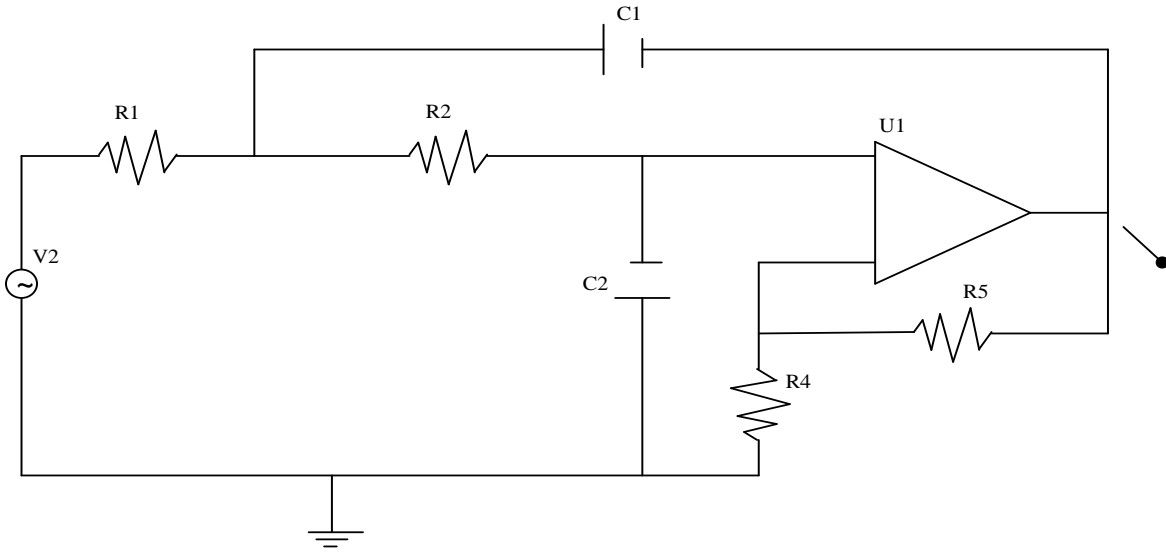

**Figure 4.** Sallen–Key low-pass filter circuit diagram.

The low-pass filter includes capacitors, resistors, and amplifiers. Because the possibility of sudden damage to these basic components is very small, in most cases in practical applications, the degradation of the components themselves causes the deviation in the actual parameter values, which in turn leads to changes in circuit performance. Through the sensitivity analyses of the circuit, it is found that the resistance R2 has the greatest influence on circuit performance. Therefore, this paper takes R2 degradation (resistance attenuation) as an example to evaluate the health status of the Sallen–Key low-pass filter.

### 3.1. Simulation

This article uses Capture in Cadence 17.4 to draw circuit schematics. The tolerances of the capacitive resistive elements are all set at 5%, the amplifier is set with no tolerance, and R2 is set as a PARAMETER element for subsequent simulations. The input signal is a 1V AC current source, and the output is the amplifier output voltage.

After the setting is completed, this paper uses Pspice in Cadence 17.4 to simulate this circuit with simulation type AC Sweep, and then the simulation settings of the following two parts are performed:

(1) General setting: The setting frequency changes in ten-fold frequency, the starting frequency is 1 Hz, the end frequency is 10 KHz, and 201 points are sampled in every ten-fold frequency.

(2) Parameter sweep setting: the parameter selected for sweep is R2, the parameter changes linearly, the initial resistance value is 32 KΩ, the end resistance value is 8 KΩ, and each change is 0.1 KΩ.

Therefore, with V(5) as the output, a total of $(32 - 8)/0.1 = 240$ sets of frequency corresponding data can be obtained, and each set of sample data samples 804 points. Figure 5 shows the frequency response curve when the resistance value of R2 is 8 K, 20 K, and 26 K. It can be seen that with the change in resistance value, the frequency response curve has changed significantly. Therefore, extracting the degradation features from the original frequency response waveform data can well characterize the degradation information of the circuit.

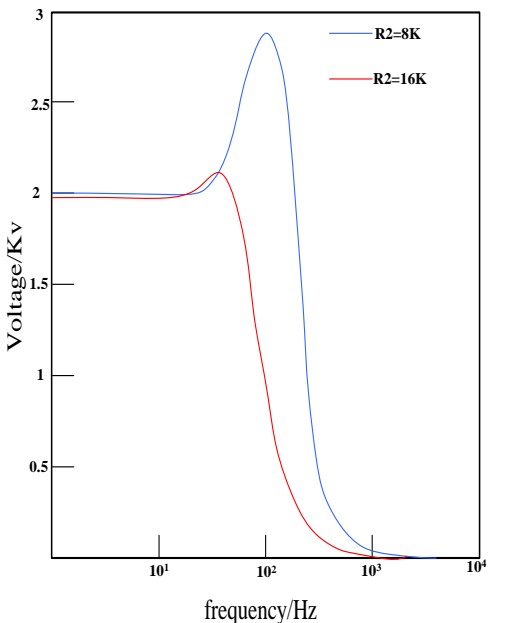
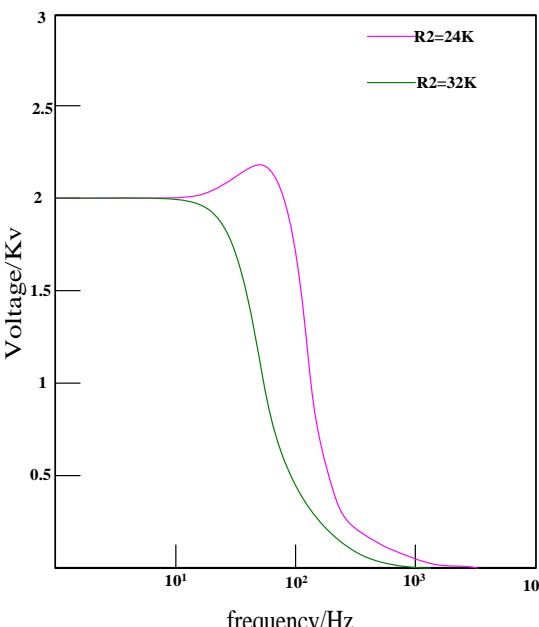

**Figure 5.** Frequency response curve under different resistance values.

### 3.2. Degradation Feature Data Extraction and Processing

The waveform data of the single frequency response curve contains a large amount of redundant data (such as low frequency area and high frequency area), which cannot be used

directly. Combined with the common characteristics of analog circuits, this paper selects the mean, variance, peak value, peak frequency, −3 dB cutoff frequency, and kurtosis. It performs feature extraction on it, and then normalizes and centralizes it after extracting features. This paper uses the mapminmax and z-score functions that come with the Matlab deep-learning tool library. After extracting and processing the feature data, some of the results obtained are shown in Table 1.

**Table 1.** Partial characteristic data.

|    | 1      | 2      | 3      | 4      | 5      | 6      |
|----|--------|--------|--------|--------|--------|--------|
| 1  | −0.401 | −0.401 | −0.402 | −0.401 | −0.401 | −0.264 |
| 2  | −0.379 | −0.398 | −0.403 | −0.405 | −0.388 | −0.269 |
| 3  | −0.394 | −0.393 | −0.404 | −0.406 | −0.391 | −0.271 |
| 4  | −0.391 | −0.389 | −0.406 | −0.407 | −0.392 | −0.281 |
| 5  | −0.379 | −0.386 | −0.411 | −0.409 | −0.393 | −0.283 |
| 6  | −0.376 | −0.385 | −0.412 | −0.412 | −0.381 | −0.291 |
| 7  | −0.381 | −0.379 | −0.416 | −0.414 | −0.382 | −0.294 |
| 8  | −0.361 | −0.376 | −0.418 | −0.415 | −0.383 | −0.298 |
| 9  | −0.379 | −0.387 | −0.429 | −0.364 | −0.379 | −0.317 |
| 10 | −0.381 | −0.386 | −0.437 | −0.354 | −0.381 | −0.319 |

In the table, columns 1–6 represent the mean, variance, peak value, peak frequency, −3 dB cutoff frequency, and kurtosis. The data bits in the table are normalized and centrally processed, and one row represents the characteristic data of a simulation test group.

Through feature extraction and processing, the change trend in each feature parameter can be obtained, as shown in Figure 6. Because the R2 set by the simulation strictly changes linearly with the number of simulations, the change in each group of features with the number of simulation groups actually changes with the change in the resistance value. Therefore, the abscissa of the figure is the resistance value. The mean value, variance, kurtosis, peak value, peak frequency, and cutoff frequency all show an increasing or decreasing trend with the change in resistance value. It is not difficult to see that each feature contains the degradation information of the Sallen–Key filter circuit. However, a single feature is still not enough to characterize the health of a circuit. Therefore, the next step is to use an improved deep-learning calculation method based on principal component weighting to fuse these features to construct a circuit health index curve.

Through statistical analyses of the performance degradation caused by component aging and parameter drift, it is generally believed that when the parameter value of the component is degraded to 25% of the original value, the impact on the performance is relatively small. Then, the degradation process is obviously accelerated. When the degradation reaches 50% of the original value, it is considered to be completely invalid and cannot be used continuously. Therefore, this paper selects the first 150 groups of sample data as the total sample set for health status assessment.

### 3.3. Calculation of Health Index

When it is considered that the initial state R2 = 32 KΩ, it is a healthy state. Therefore, the first 20 groups are used as healthy state samples, and 150 groups of samples are used as test samples, and the depth distance between each group of data and healthy state samples is calculated as the health factor of the Sallen–Key low-pass filter. Because the calculated distance may be too large in magnitude, it is necessary to perform appropriate scaling to obtain the corresponding representative health index. The mean values of the parameters of the healthy state samples are shown in Table 2, and the covariance matrix is shown in Table 3.

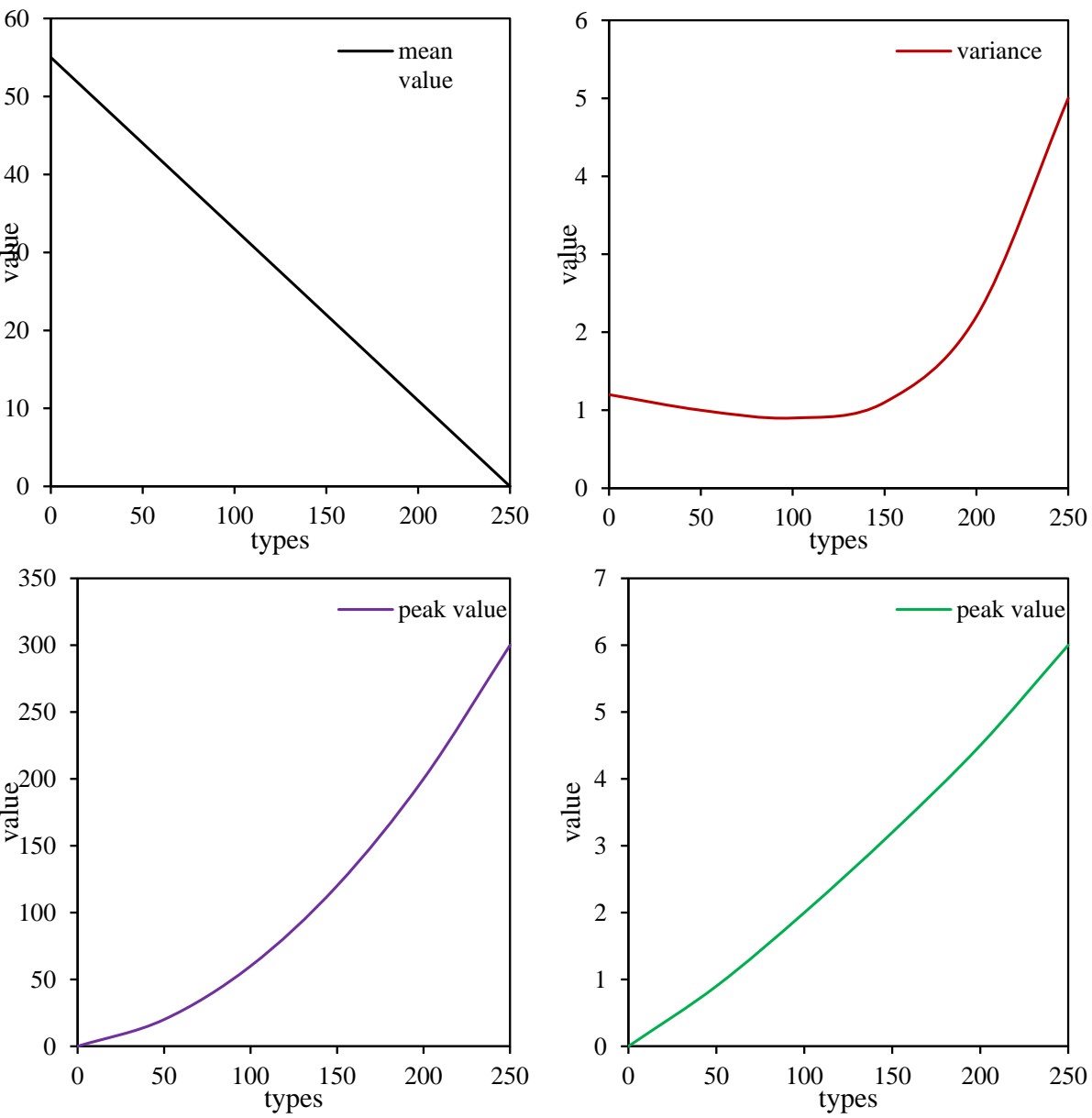

**Figure 6.** Partial characteristic change curve.

**Table 2.** Sample mean data.

|  | **1** |  | **1** |
| --- | --- | --- | --- |
| 1 | −0.379 | 4 | −0.319 |
| 2 | −0.389 | 5 | −0.401 |
| 3 | −0.439 | 6 | −0.329 |

Because the filter circuit here has no nominal value, any set of health data within 5% of the tolerance is used as the new mean. The specific values of the new mean matrix used in this paper are shown in Table 4, and then the depth distance based on principal component weighting is calculated. The calculation results of the weight matrix are shown in Table 5.

**Table 3.** Sample covariance matrix data.

|  | 1 | 2 | 3 | 4 | 5 | 6 |
|---|---|---|---|---|---|---|
| 1 | $6.239 \times 10^{-5}$ | $3.729 \times 10^{-5}$ | $-1.529 \times 10^{-4}$ | $2.855 \times 10^{-4}$ | $9.901 \times 10^{-6}$ | $-2.345 \times 10^{-4}$ |
| 2 | $3.729 \times 10^{-5}$ | $3.441 \times 10^{-5}$ | $1.181 \times 10^{-5}$ | $-1.149 \times 10^{-4}$ | $2.619 \times 10^{-5}$ | $1.261 \times 10^{-6}$ |
| 3 | $-1.529 \times 10^{-2}$ | $1.178 \times 10^{-5}$ | $0.0012$ | $-0.0029$ | $1.502 \times 10^{-4}$ | $0.0021$ |
| 4 | $2.849 \times 10^{-4}$ | $-1.149 \times 10^{-4}$ | $-0.0029$ | $0.0079$ | $-4.661 \times 10^{-4}$ | $-0.0039$ |
| 5 | $9.91 \times 10^{-6}$ | $2.619 \times 10^{-5}$ | $1.501 \times 10^{-4}$ | $-4.661 \times 10^{-4}$ | $6.479 \times 10^{-5}$ | $2.001 \times 10^{-4}$ |
| 6 | $-2.35 \times 10^{-4}$ | $1.261 \times 10^{-6}$ | $0.0017$ | $0.0039$ | $2.001 \times 10^{-4}$ | $0.0024$ |

**Table 4.** Improved mean data.

|  | 1 |  | 1 |
|---|---|---|---|
| 1 | $-0.378$ | 4 | $-0.415$ |
| 2 | $-0.382$ | 5 | $-0.379$ |
| 3 | $-0.416$ | 6 | $-0.294$ |

**Table 5.** Weight matrix data.

|  | 1 | 2 | 3 | 4 | 5 | 6 |
|---|---|---|---|---|---|---|
| 1 | $0.0049$ | $0.0029$ | $-0.0131$ | $0.0241$ | $8.1901 \times 10^{-4}$ | $-0.0202$ |
| 2 | $0.0029$ | $0.0031$ | $9.751 \times 10^{-4}$ | $-0.0089$ | $0.0019$ | $1.039 \times 10^{-4}$ |
| 3 | $-0.0131$ | $9.681 \times 10^{-4}$ | $0.1051$ | $-0.3001$ | $0.0109$ | $0.1479$ |
| 4 | $0.0241$ | $-0.0101$ | $-0.2599$ | $0.7002$ | $-0.0401$ | $-0.3659$ |
| 5 | $8.191 \times 10^{-5}$ | $0.0019$ | $0.0099$ | $-0.0401$ | $0.0049$ | $0.0201$ |
| 6 | $-0.0201$ | $1.039 \times 10^{-4}$ | $0.1501$ | $-0.4003$ | $0.0201$ | $0.2099$ |

The health value evaluation of the final Sallen–Key low-pass filter is shown in Figure 7:

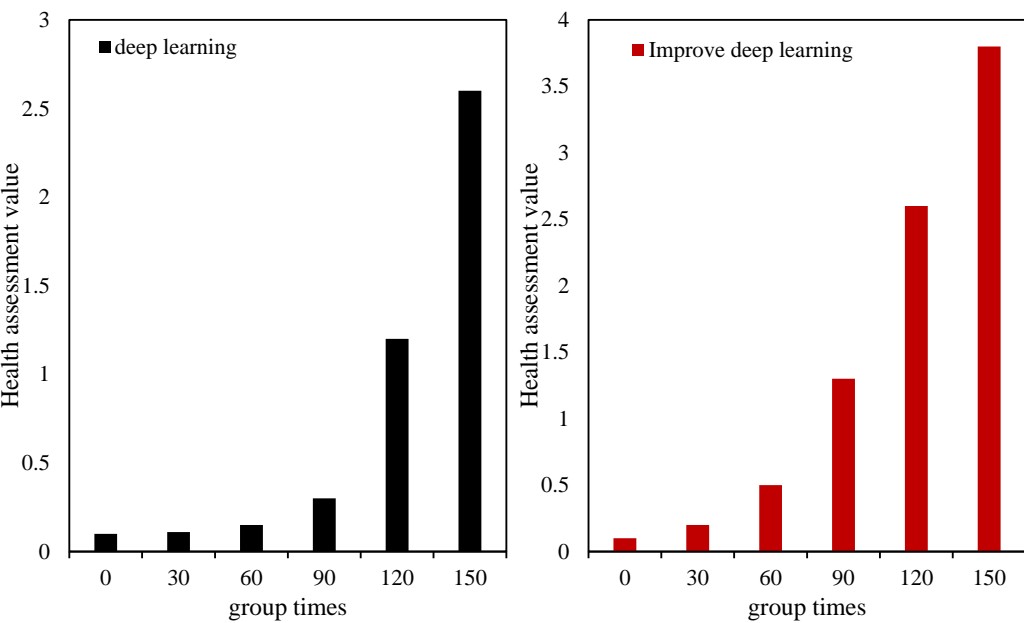

**Figure 7.** Health index assessment.

*3.4. Results*

The preset performance change point of the Sallen–Key low-pass filter simulated in this paper is when the resistance value of R2 degenerates to 1/4 of the original resistance value, that is, when it is 24 KΩ. The performance failure point is when the resistance value is 16 KΩ. Therefore, it can be considered that the effective use time of the entire filter circuit is the time when the resistance value changes to 16 KΩ. Among them, for the entire life cycle of the circuit, the value interval of the health index curve based on improved deep learning is [0–3.8]. The health value range of unimproved deep learning is [0–2.6]. Compared with the two, the range of health values for the whole lifespan is expanded by about 1.5 times with improved deep learning. This also means the health index curve covers a wider range and can more intuitively display the trend in the health index.

Before the performance change point, the change in unimproved deep learning was small, while the improvement in deep learning increased the health value by nearly 10 times. Additionally, the degradation interval of improved deep learning is almost equal to the original degradation interval. Improved deep learning thus amplifies subtle changes in pre-degeneration health. It slows down the mutation of health in late stages near the point of performance failure. Figure 8 shows the comparison of the health-index-evaluation indicators constructed by the two methods. It can be seen that although the monotonicity of the health index is slightly reduced, its robustness and correlation are significantly improved and are very close to 1. This makes the health index curve more in line with conventional cognition and is convenient for practical operations. Therefore, the method of health assessment based on improved deep learning has practical significance.

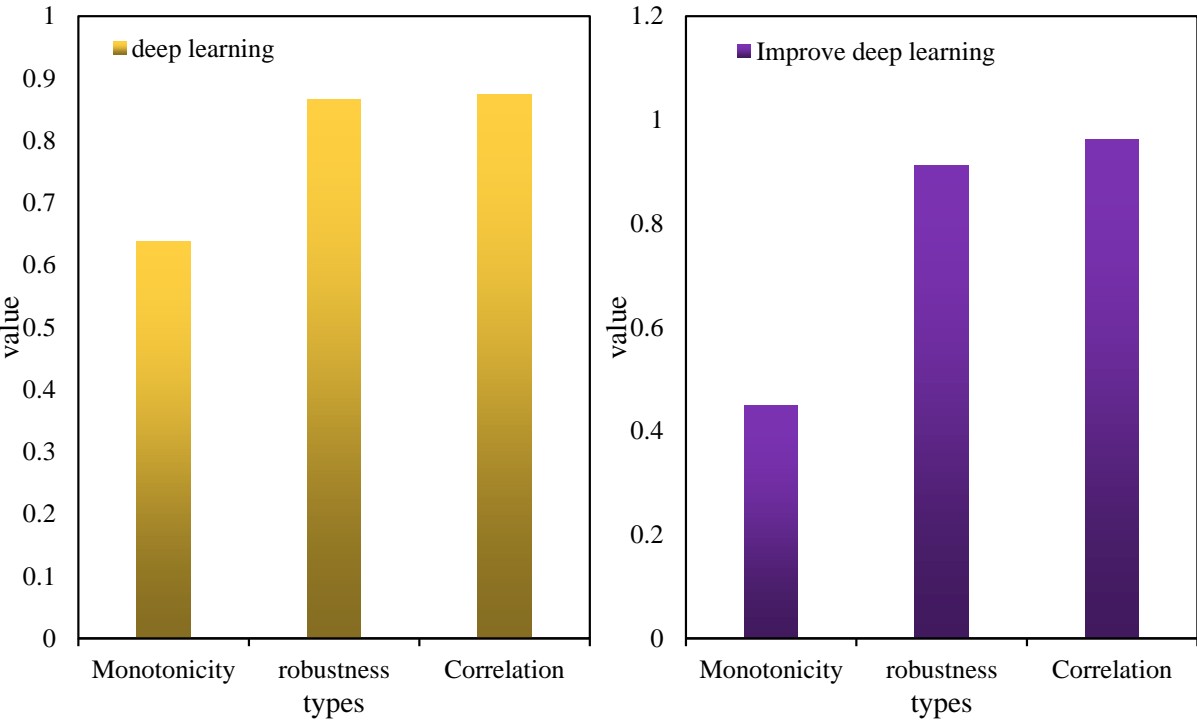

**Figure 8.** Results of health index evaluation constructed by two methods.

## 4. Conclusions

The importance of failure prediction and health management (PHM) to complex equipment or systems cannot be overemphasized. This is especially true for today's electronic equipment, which is highly integrated and modular, meaning the complex equipment/system architecture often have distinct layers and independent functions. Each part has a specific function and role. Therefore, if a problem occurs in a certain part, it affects the whole body and directly affects the performance of the entire equipment/system.

This paper firstly summarizes the origin, significance, and research situation of health status assessment, and then classifies them according to the compositional properties of intelligent management and control systems. This paper summarizes the main process of health status assessment in the predictive maintenance of the intelligent management and control system. Then, this paper gives three common evaluation indicators for health status evaluation: monotonicity; robustness; and correlation. These are used to determine whether the health index curve is good or not. In this paper, a health-state-assessment method based on improved deep learning is proposed. In view of the problems existing in traditional deep learning, two improvements have been made to deep learning. It uses improved deep learning to calculate and construct a health index curve to represent health status. At last, the simulation experiment is carried out on the Sallen–Key low-pass filter circuit. The experimental results show that the health assessment method based on improved deep learning can clearly represent the changes in health status. It has practical use significance and can provide technical support for the health management of the intelligent management and control system.

**Author Contributions:** P.W., J.Q. and J.L. designed and performed the experiment and prepared this manuscript. M.W., S.Z. and L.F. helped conduct the experiment. All coauthors contributed to manuscript editing. All authors have read and agreed to the published version of the manuscript.

**Funding:** This work was supported by the National Key R&D Program of China (No. 2019YFB1600400).

**Institutional Review Board Statement:** Not applicable.

**Informed Consent Statement:** Not applicable.

**Data Availability Statement:** The data that support the findings of this study are available from the corresponding author upon reasonable request.

**Conflicts of Interest:** The authors declare no conflict of interest.

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
