# Peer review of "Device Status Evaluation Method Based on Deep Learning for PHM Scenarios"

_electronics, doi:10.3390/electronics12030779_

Round 1

Reviewer 1 Report

I really appreciate this paper, which has carried out in-depth analysis and research on fault prediction and health management. The framework of the paper is very clear, the idea is very clear, and the direction and content of the research are also very clear. Of course, there are also some defects that need to be improved.

1Health assessment is the key link of predictive maintenance. It plays a vital role in the PHM model of the entire intelligent management and control system. Which aspects do the research work of health status assessment mainly focus on?

2According to the different objects controlled by the intelligent management and control system, the health status evaluation process is slightly different. However, for complex equipment/systems, they can be roughly divided into two types according to their composition: hierarchical structure and non hierarchical structure. What are the common health assessment methods?

3Based on the theory of deep learning, a deep neural network model of deep learning is constructed. The model extracts, processes and blurs information through neurons in each layer. In which areas has deep learning achieved a lot?

4In the whole learning process, the network model can be self-learning based on a large number of data information. The distribution characteristics and complex function representation of learning information are stored in the deep neural network, so that it has the ability of recognition and perception. Where does the concept of deep learning come from?

5The learning ability of deep neural network is embodied in the expression of weight and deviation of each neuron, and is used to fit a strong mapping relationship. What is the difference between shallow learning and deep learning?

6Back propagation is to transfer the error signal from the output layer to the input layer layer by layer, and adjust the weight coefficient in the transmission process to make the network output reach the expected output. What is the basis of the back propagation algorithm?

Author Response

I really appreciate this paper, which has carried out in-depth analysis and research on fault prediction and health management. The framework of the paper is very clear, the idea is very clear, and the direction and content of the research are also very clear. Of course, there are also some defects that need to be improved.

1Health assessment is the key link of predictive maintenance. It plays a vital role in the PHM model of the entire intelligent management and control system. Which aspects do the research work of health status assessment mainly focus on?

 A: Health status assessment methods The research work of health status assessment is mainly focused on the assessment methods, which are mainly based on the characteristics of specific research objects and various assessment methods.

2According to the different objects controlled by the intelligent management and control system, the health status evaluation process is slightly different. However, for complex equipment/systems, they can be roughly divided into two types according to their composition: hierarchical structure and non hierarchical structure. What are the common health assessment methods?

 A: Common health assessment methods include: model method, analytic hierarchy process, fuzzy evaluation method, artificial neural network method, Bayesian network-based method, grey theory, extension theory, etc.

3Based on the theory of deep learning, a deep neural network model of deep learning is constructed. The model extracts, processes and blurs information through neurons in each layer. In which areas has deep learning achieved a lot?

 A: Deep learning has made many achievements in search technology, data mining, machine learning, machine translation, natural language processing, multimedia learning, voice, recommendation and personalized technology, and other related fields.

4In the whole learning process, the network model can be self-learning based on a large number of data information. The distribution characteristics and complex function representation of learning information are stored in the deep neural network, so that it has the ability of recognition and perception. Where does the concept of deep learning come from?

 A: The concept of deep learning originates from the study of artificial neural networks. Multilayer perceptron with multiple hidden layers is a good example of deep learning model.

5The learning ability of deep neural network is embodied in the expression of weight and deviation of each neuron, and is used to fit a strong mapping relationship. What is the difference between shallow learning and deep learning?

 Answer: Different from the traditional shallow learning, the difference of deep learning is that it emphasizes the depth of the model structure. There are usually 5, 6, or even 10 layers of hidden nodes; It clearly highlights the importance of feature learning, that is, through layer-by-layer feature transformation, the feature representation of the sample in the original space is transformed into a new feature space, which makes classification or prediction easier.

6Back propagation is to transfer the error signal from the output layer to the input layer layer by layer, and adjust the weight coefficient in the transmission process to make the network output reach the expected output. What is the basis of the back propagation algorithm?

A: Back propagation algorithm, or BP algorithm for short, is a learning algorithm suitable for multi-layer neural networks. It is based on gradient descent method.

Reviewer 2 Report

I congratulate the author on this fine work,

Below are some major notes on the manuscript:- 

1-    Readers use the abstract to quickly learn the topic of your research. A well-written summary is critical to attracting readers so that they can open up and read about your work.
So, abstracts should highlight major points of your research and should explain why your work is important; what your purpose was.

2-    There is a mixture between the literature and the introduction, Preferably the introduction serves the purpose of leading the reader from a general subject area to a particular field of research. the introduction should introduce your topic and aims and gives an overview of the paper.

3-    Line 92-93 should be referenced.

4-    The author should state an argument for why the extracting feature is an extremely challenging as mentioned in line 175, that’s make the author go through simple feature extraction.

5-    The author should add a section for Dataset or data source description, a description of collected data set volume and its structure helps readers to easier understand the data

6-    It is much better to show the result of neural network performance such as validation loss and the accuracy of testing the model, recall, precision, and F1 score.

7- The author should make sure that volume, pagination, and DIO number are mentioned in the reference's bibliography. 

Author Response

I congratulate the author on this fine work,

Below are some major notes on the manuscript:- 

1-    Readers use the abstract to quickly learn the topic of your research. A well-written summary is critical to attracting readers so that they can open up and read about your work.

So, abstracts should highlight major points of your research and should explain why your work is important; what your purpose was.

 A: In order to solve such problems, this paper studies the application of equipment status assessment method based on deep learning in PHM scenarios, with a view to conducting in-depth research on the intelligent control system of electronic equipment.

2-    There is a mixture between the literature and the introduction, Preferably the introduction serves the purpose of leading the reader from a general subject area to a particular field of research. the introduction should introduce your topic and aims and gives an overview of the paper.

 A: This paper first analyzes the PHM health status assessment process and introduces the relevant indicators of health status assessment. A health state assessment method based on improved deep learning is proposed. Finally, the health assessment of PHM model is analyzed through simulation experiments. It aims to make certain contributions to the health assessment of the PHM model.

3-    Line 92-93 should be referenced.

 A: Thank you for your suggestion. We have already referred to it.

4-    The author should state an argument for why the extracting feature is an extremely challenging as mentioned in line 175, that’s make the author go through simple feature extraction.

 Answer: In machine learning, pattern recognition and image processing, feature extraction begins with an initial set of measurement data, and establishes derived values (features) aimed at providing information and non-redundancy, thus facilitating the subsequent learning and generalization steps, and in some cases, bringing better interpretability.

5-    The author should add a section for Dataset or data source description, a description of collected data set volume and its structure helps readers to easier understand the data

 A: Thank you for your suggestion. We have dealt with it.

6-    It is much better to show the result of neural network performance such as validation loss and the accuracy of testing the model, recall, precision, and F1 score.

 Answer: The accuracy rate and the recall rate are contradictory. The recall rate is often very low when the model has a high accuracy rate. On the contrary, the higher the recall rate is, the lower the accuracy rate is. Therefore, in practical application, we often need to consider the accuracy rate, accuracy and recall rate as a whole according to the actual needs, so as to train the optimal model.

  • The author should make sure that volume, pagination, and DIO number are mentioned in the reference's bibliography. 

A: Thank you for your suggestion. We have supplemented the references of the article.

Round 2

Reviewer 2 Report

Thanks for your replies, and corrected notes.

Good luck.